# JPEG Images Encryption Scheme Using Elliptic Curves and A New S-Box Generated by Chaos

Erick Hernández-Díaz [1], Héctor Pérez-Meana [1,*], Víctor Silva-García [2] and Rolando Flores-Carapia [2]

[1] SEPI ESIME Culhuacan, Instituto Politécnico Nacional, Mexico City 04400, Mexico; erick_hd@live.com.mx
[2] CIDETEC, Instituto Politécnico Nacional, Mexico City 07700, Mexico; vsilvag@ipn.mx (V.S.-G.); rfloresca@ipn.mx (R.F.-C.)
* Correspondence: hmperezm@ipn.mx

**Abstract:** This paper proposes a new symmetric encryption system based on an elliptical curve and chaos, where the encryption is done in a single block and runs for 14 rounds. Here, the 15 encryption keys have the same size as the image and are generated using a solution point of a strong elliptic curve. Using a string of random numbers obtained with a logistic map, a permutation and its inverse are generated, which improve the encryption level and add diffusion to the cryptosystem. Another important contribution to this research is the generation of a substitution box with a non-linearity of 100, which strengthens the cryptosystem against differential and linear attacks that add confusion to the scheme. Moreover, the cryptographic properties of the proposed S-Box are compared with those of the S-Box of the Advanced Encryption Standard (AES) to ascertain that it is a suitable alternative that it is resistant to differential power analysis (DPA) attacks. To verify the robustness of proposed cryptosystem against cryptanalysis and the quality of the cipher text obtained, this system is subjected to different tests, such as entropy, correlation coefficient, $\chi^2$, Changing Pixel Rate (NPCR), and Unified Averaged Changing Intensity (UACI). The results are shown and compared with recently proposed systems.

**Keywords:** applied cryptography; images encryption; jpeg encryption; elliptic-curve cryptography; chaos; logistic map; cryptanalysis; randomness; symmetric cryptosystem; substitution box

## 1. Introduction

A large amount of sensitive information is transmitted through the internet, and such information can be easily accessed through public as well as private networks. The images, which represent an important amount of transmitted information, can be photographs, agreements, contracts, identification documents, account statements, or other kind of scanned documents, with a high intrinsic value. Any person with a mobile device and an internet connection will be able to send images through the internet and store them in physical devices or in the cloud. The security and confidentiality of such information have led to the development of several efficient cryptographic systems.

In some practical applications, involving scientific and engineering fields such as astronomy or medicine, the use of images without compression and loss information is commonly required. Among the image formats that meet these requirements, we have the Microsoft-designed Bitmap (BMP) format, which contains all the image information in a simple format operating in the spatial domain. It can be modified and easily edited because it can be debugged and viewed without special tools; besides that, it is used to display image files, including the color of each pixel, and although it was designed to be a Windows standard, it is currently supported by a variety of operating systems [1]. In several countries, such as Mexico, where this research was carried out, it is prohibited to lose information in documents with sensitive information that need to be encrypted [2]; the use of BMP or another lossless scheme is imperative when it is required to store digital documents. To encrypt the BMP images, reference [3] proposed an efficient cryptographic

algorithm based on elliptic curves and chaos. However, the size of BMP format images is too large for many applications; then, it is necessary to use some compressing format. Thus, this paper proposes an encryption algorithm for use in JPEG (Joint Photographic Experts Group) format [4], which is widely used in a variety of electronic devices such as smartphones, computers, or digital cameras. This task is carried out in two stages: the baseline, which has no loss of information, and DCT (Discrete Cosine Transform), which considerably reduces the file size due to the loss of information, which is achieved by discarding redundant pixels using a quantization matrix, reducing in such way the image size and doing it in an easier manner than file transmission or storing [4]. It is important to mention that the picture quality will depend on the compression level; usually, a level from 95% to 0.70% will give excellent quality but low compression, while if the maximum level is applied, 0% to 20%, the resulting image quality will degrade and sometimes become indistinguishable. Moreover, most of the other image formats are based on the RGB color models (Red, Green, Blue), but the JPEG format employs the color model YUV, where "Y" means luminance while "U" and "V" means chrominance, which works under the principle that the human eye is unable to identify certain bands of colors [5]. Thus, although there is a loss of information, it is imperceptible to the human eye.

Protecting images with sensitive information transmitted through public communication channels is an important task that has been a topic of active research during the last several decades, leading to the development of several efficient cryptographic algorithms. To this end, recently, researchers have developed some efficient cryptosystems based on chaotic systems and/or in elliptic curves that avoid discrete logarithm attack, since it has been proven that they are robust against differential [6,7], linear [8], or even statistical cryptanalysis.

Several efficient cryptographic schemes have been proposed recently in the literature based on elliptic curves and chaos, such as those described below. In [9], the authors propose two image encryption schemes in which their encryption keys are generated using the logistic equation that is presented in Equation (13) and Henon maps. Then, the image encryption is performed using the solution points of an elliptic curve and applying bitwise XOR operations in three different stages. In [10], the authors propose a three-step scheme where first, it uses two chaotic maps known as LTM and TSM to permute the pixels of the input image; the input parameters of these are obtained using the SHA-512 of the input image, from which the encryption keys are also generated. Then, using an asymmetric cryptosystem known as Elliptic Curves-ElGamal, the image is encrypted. Finally, using a genetic algorithm based on DNA sequences, diffusion is applied to the previously obtained output. In [11], the authors propose an encryption scheme in two scenarios; both use an Elliptic Curves-Diffie-Hellman protocol and another key generated from the SHA-256 of the input image. Furthermore, in the encoding process, a triple chaotic STH map is used, and a digital signature is added to verify its origin before being decrypted. In [12], the authors propose a scheme that uses ECC to share a point of an elliptic curve that will serve to obtain the initial parameters $(x_0, \mu)$ of a logistic map and the number of rounds of Arnold's map that will serve to permute the pixels from an image. The parameter k of the elliptic curve used is obtained with the SHA-512 and using the multiplication of points between k and a generator, the encryption keys are created. In [13], the authors propose a four-step scheme. Here, firstly, they reduce the input image, which in this case is an RGB image, and extend it in the grayscale for the entire field E (F_p). Secondly, they apply an enhanced 4D Arnold cat map t times to permute the image pixels. Next, they encrypt the image with an ECC. Finally, they apply a 3D Lorenz chaotic map to add diffusion to the final encrypted image. Finally, in [14], the authors propose a new method called EC-GRP permutation operation, which combines the cryptographic properties of the ECC and the operation group. This procedure requires a pseudo-random bit generator with good attributes according to the cryptography field, which is used to encrypt the input image. According to the researchers, the results obtained are highly resistant to cryptanalysis tests.

Despite the efficient algorithms described above, several issues remain that must be improved. To this end, this paper proposes a cryptographic algorithm for improving the above-mentioned schemes, whose main contributions are summarized as follows: (a) It uses elliptic curves with a constant *l* equal to zero for the generation of the set of encryption keys. (b) It proposes an algorithm to generate the elliptic curves in addition to being mandatory that they comply with certain characteristics. (c) It uses a chaotic logistic equation to generate permutations and a substitution box plus its inverse with a non-linearity level of 100. (d) It implements a fingerprint for the receiver to identify if the received file corresponds to the one sent by the issuer. (e) It encrypts the images in a single block, which allows obtaining an adequate encryption speed. On the other hand, the cryptosystem was evaluated using several tests. After analyzing the evaluation of the results reported in Section 5, it follows that the proposed structure is robust and capable of withstanding linear, differential, statistical, brute force, or modification attacks, as well as some of the better known as the discrete logarithm and the MOV.

The rest of this paper is organized as follows. In Section 2, the preliminaries of this research are stated. Section 3 describes how the encryption keys, the permutations, the substitution box, and the complete encryption algorithm are developed. Section 4 provides several essential information such as the images that are encrypted, the explanation of every test necessary to demonstrate how robust is our proposal against cryptanalysis, and the results obtained. In Section 5, the analysis of the data obtained after applying all the tests is provided. Finally, Section 6 presents the conclusions of this research.

## 2. Preliminaries

This section provides a background and a detailed explanation about the two pillars on which this research is based: the elliptic curves and chaos.

### 2.1. Elliptic Curves in the Field of Cryptography

The elliptic curves have been used in many fields of science and engineering since their discovery. However, they were firstly introduced in the cryptography field in 1985 and 1987 by the mathematicians, Victor S. Miller in [15] and N. Koblitz [16]. An elliptic curve $E$ is a projective geometric shape that is defined in a field $F_p$ in which a set of solutions $\#E(F_p)$ is constructed that has two variables that satisfy the Weierstrass' mathematical expression given by

$$y^2 \equiv x^3 + kx + l \ mod \ p. \tag{1}$$

This set of solutions is Abelian, in which it is possible to define the addition operation $(E, +)$. In $\#E(F_p)$, a cyclic subset is built, where $q$ is a prime factor that represents the number of elements (solutions) of this and the order of the elliptic curve. The prime factor q is calculated as follows:

$$q = \frac{p + 1 + 2a}{4} \ mod \ p. \tag{2}$$

The element of the subset that generates all other solutions is known as the generator and is denoted with the Greek letter $\alpha$. The element $(q - 1)\alpha$ is the additive inverse of $\alpha$; that satisfies $(q - 1)\alpha = (x_0, -y_0)$. So, $q\alpha$ is the null element, which is written as infinity $\infty$ [17]. As it has already been established, with the addition of the other elements, it can be calculated—that is $2\alpha$, $3\alpha$. To calculate their coordinates $(x, y)$, it is necessary to obtain the slope of the line and there can be three cases, which are described below.

a.      When $x_0 \neq x_1$, Equations (3)–(5) are used.

$$\lambda = (y_1 - y_0)(x_1 - x_0)^{-1} \ mod \ p, \tag{3}$$

$$x_2 = \left(y^2 - x_0 - x_1\right) \ mod \ p, \tag{4}$$

$$y_2 = \lambda(x_0 - x_2) - y_0 \ mod \ p. \tag{5}$$

b.　　When $x_0 = x_1$ and, $y_0 = y_1$, Equations (3)–(5) are used.

$$\lambda = \left(3x_0^2 + a\right)(2y_0)^{-1} \ mod \ p,\tag{6}$$

$$x_2 = \left(\lambda^2 - 2x_0\right) \ mod \ p.\tag{7}$$

c.　　When $x_0 = x_1$ and $y_0 = -y_1$, the null element is obtained, such that $(x_0, y_0) + (x_0, -y_0) = \infty$, it means that both are inverses concerning the elliptic curve addition operation. Furthermore, $\alpha + \infty = \alpha$.

In this research, we use the elliptic curves that satisfy Equation (8), whose palpable difference with (1) is that the constant $l$ is equal to zero; thus, for practical reasons, it is omitted in the mathematical expression,

$$y^2 \equiv x^3 - kx \ mod \ p.\tag{8}$$

The elements $k$, $p$, $q$ are prime numbers; to ensure that they have this characteristic, it is necessary to apply a primality test such as Miller Rabin's [18]. To calculate $k$, Equation (9) is used in addition to complying with the characteristics described in Theorem 1 [19], which is also useful to explain how $\#E(F_p)$ is calculated, that is

$$k = \left(x_0^3 - y_0^2\right)(x_0)^{-1} \ mod \ p.\tag{9}$$

**Theorem 1.** *Let $p = a^2 + b^2$ and $p \equiv 1 \ mod \ 4$; where $p$ is a prime number, $a$ a positive odd integer, $b$ a positive even integer, and $\#E(F_p)$ the number of solutions of the elliptic curve shown in (8). Then, the number of solutions is $\#E(F_p) = p + 1 + 2a$, provided that $k$ is not the fourth power modulo $p$ of any element of the field $F_p$; however, $k$ must be the power squared of some element of the field $F_p$ [19].*

In addition to having this expression, they must meet four requirements:

a.　　Being Non-Singular [17]. This happens when an elliptic curve meets the condition: $4(-k)^3 \ mod \ p \ 0$.

b.　　To be Non-Supersingular [17]. Non-Supersingular elliptic curves resist the attack designed by Menezes, Okamoto, and Vanstone [20]. They fulfill the following condition: $q \ mod \ p \ 1$.

c.　　Not be an Elliptic Curve of Trace One [17]. The elliptic curves of Trace One are weak. If they do not belong to this type, they must satisfy that $p \neq q$.

d.　　The size of q is at least $2^{160}$ bits [17]. With this characteristic in the elliptic curves, it is impossible to solve the problem of the discrete logarithm that is capable of finding $m$ with $Q = mP$ when $P$ and $Q$ are known [6]. Even though the existence of the Pohlig–Hellman algorithm can solve this problem [21], it requires a high computational cost and exponential processing time.

There are research works that show that the elliptic curves such as those described in this paper are robust and adequate for the development of safe systems [3–22]. Figure 1 presents the complete algorithm for the search of the elliptic curves to be used in this research.

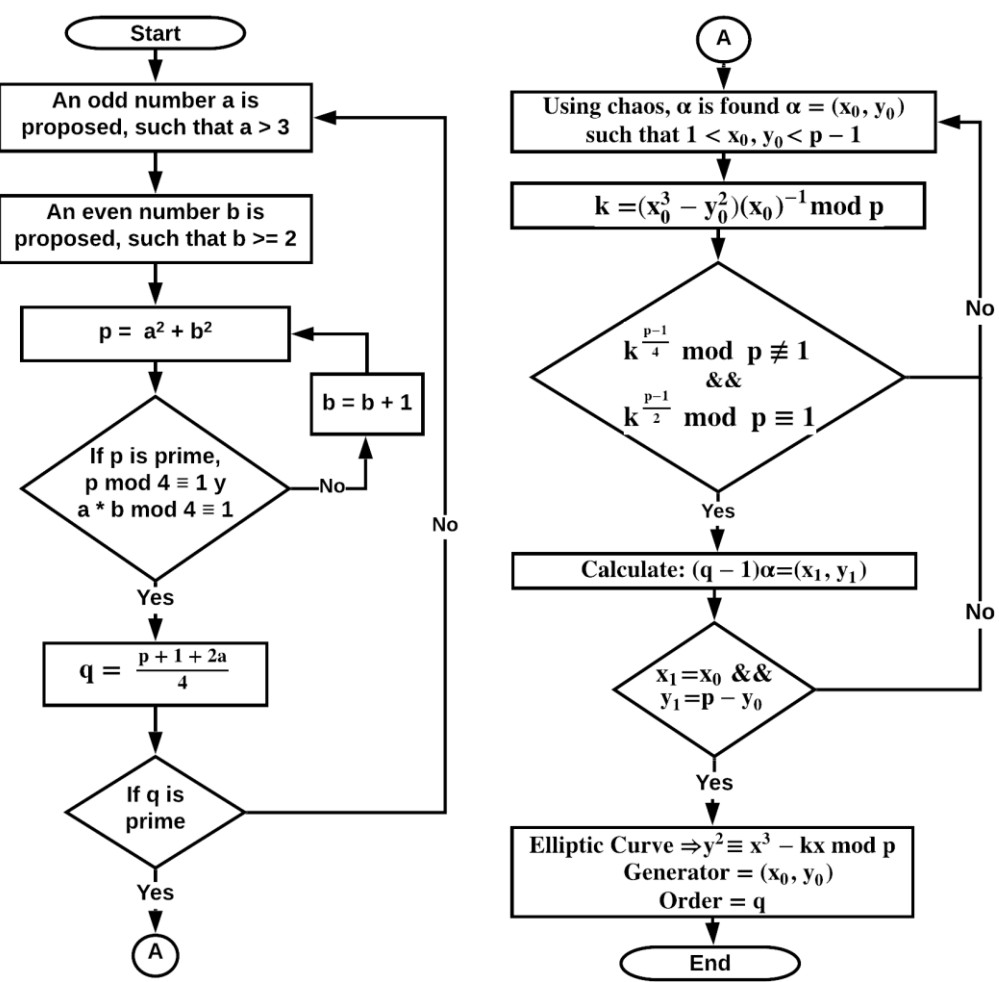

**Figure 1.** Flow diagram that describes the algorithm for the search for the elliptic curves used in this research.

### 2.2. Logistic Map

The Australian mathematician Robert May [18] proposed a logistic map to explain the abnormal behavior in the growth of populations; however, in the field of Cryptography, it is very useful to generate strings of pseudo-random numbers. This equation is obtained from the discrete expression shown in Equation (10), where $e, f > 0$.

$$\frac{dP}{dt} = eP - fP^2, \tag{10}$$

After this, the variable $P$ takes discrete values in time such that $P(t_0), \ P(t_1), P(t_2) \ldots, P(t_n)$. Afterward, Euler's Algorithm is applied so that the result obtained is shown in Equation (11) [23].

$$P_{n+1} = P_n + \left(eP_n - fP_n^2\right) * h. \tag{11}$$

The next step is to rewrite the equation as shown in Equation (12), where $s = 1 + eh$ and, $t = fh$.

$$P_{n+1} = sP_n - tP_n^2. \tag{12}$$

Lastly, the iterative equation shown in Equation (13) is obtained. Where $0 < x_n < 1$ and $0 < s < 4$ [24],

$$x_{n+1} = sx_n(1 - x_n), \tag{13}$$

where $0 < x_n < 1$ and $0 < s < 4$ [24]. Two behaviors can be observed when iterating the logistic equation after 1000 cycles: a limit with a tendency to infinity ($x_\infty = \lim_{n \to \infty} X_n$),

in which case it is stated that chaos occurs, or stabilization of the digits at the right of the decimal point, which implies that chaos does not take place, since there is a pattern to follow. Both are exemplified in Table 1.

**Table 1.** Examples when chaos is done or not after 1000 iterations.

| $x$ | $s$ | Result | Observation |
|---|---|---|---|
| 0.01 | 3.1097 | 0.5528279577750637 | Chaos occurs. |
| 0.01 | 3.1098 | 0.55277575120222222 | Chaos does not take place. |
| 0.001 | 2.475 | 0.595959595959595 | Chaos does not take place. |
| 0.001 | 2.476 | 0.5961227786752827 | Chaos occurs. |

This logistic equation has three main characteristics:

a.   It is deterministic; if the initial parameters are the same, the same string will be obtained.

b.   It is highly sensitive to changes, as seen in Table 1; by varying one of the parameters a little, the chain obtained is different.

c.   The output string is impossible to predict, since it is not the result of any algebraic equation with rational coefficients, unless the input parameters are known beforehand, in which case it might be found.

## 3. Development

In this section, we provide a complete description of how the proposed cryptosystem is developed. Firstly, the algorithm used to generate the rounds keys is explained, which is followed by the way to get permutations and the proposed S-Box, and finally, all the previous pieces are joined to explain the complete encryption scheme.

### 3.1. Encryption Keys

The security of any cryptosystem relies on the encryption key; anyone with access to it will be able to decrypt the information sent. In this study, a private key cryptosystem (also called symmetric) is proposed in which the main key is private; that is, it will be exclusively known to the sender and receiver. From this one and using a keys schedule, all the round keys are generated. In this type of scheme, the set of encryption keys are the same for the encryption and the decryption process, but to decrypt, they are employed strictly in reverse order.

The proposed symmetric cryptosystem generates the main key $K$ from a point of an elliptic curve that meets the characteristics described in section II and that must be known to all the people involved in the secure communication scheme. In total, 15 round keys are required.

To begin with, the sender and the receiver will choose an integer $r$, which is the private key, and which will refer to a solution point of an elliptic curve; also, it will have to fulfill with $1 < r < p - 1$, and its size in the binary expression must be equal to 256 bits. From $r$, a new subset solution will be constructed, that is, $r\alpha = (x_0, y_0) \therefore \alpha = (x_0, y_0)$. Starting from $\propto$, all the coordinates $x$, $y$ of every point will be concatenated to get a string of the same size as the input image $(m \times n)$; if its length is overcome, it will be adjusted suppressing the surplus bits. For example, if the input image has the dimensions $512 \times 512$, that is, 262,144 pixels, the length of the array $K$ must be 262,144 bits after any adaptation.

$$K = \alpha \,||\, 2\alpha \,||\, 3\alpha \,||\, \ldots \,||\, n\alpha \,\therefore\, K = x_0 \,||\, y_0 \,||\, x_1 \,||\, y_1 \,||\, x_2 \,||\, y_2 \,||\, \ldots \,||\, x_n \, y_n \qquad (14)$$

Once $K$ has been obtained, the procedure to obtain the first round key $k_1$ will be to permute the bits of each key with the S-Box that is presented in Section 3.3 to increase its level of randomness; then, at the output obtained, a 5-bit shift is applied to the right, and

this process will be repeated to obtain $k_2$ but this time starting from $k_1$, and so on until leading to $k_{15}$ whose origin will be $k_{14}$.

$$k_1 = (S_{BOX}(K)) \gg 5;$$

$$k_2 = (S_{BOX}(K_1)) \gg 5;$$

$$\dots\dots\dots\dots\dots$$

$$k_{15} = (S_{BOX}(K_{14})) \gg 5$$

Since each coordinate of a point is considered as pseudo-random, $K$ is considered as a pseudo-random string, too. The only way an attacker could infer $K$ is to know the value $r$ and the elliptic curve chosen to start the concatenation of values, which can be solved by protecting the process of sending this information through insecure means by encrypting it with some asymmetric cryptosystem. In addition, it would face the problem that an elliptic curve similar to the one proposed in this research and presented in Section 4.2 can have a solution set with an extension of at least $2^{160}$ [12].

*3.2. Permutations and the Proposed Substitution Box*

One of the objectives of any cryptosystem is to obtain an encrypted text with a high level of randomness; this characteristic ensures that a scheme is resistant to differential and linear cryptanalysis. Having said the above, two techniques help to strengthen any encryption system: these are diffusion and confusion [25]. Diffusion disperses the elements of the plain text so that one may hide the relationship that exists between it and the ciphertext obtained at the output; to achieve it, permutations are used. The confusion confuses an attacker and makes it difficult for him to establish a relationship between the ciphertext and the key. Substitution boxes (S-Box) are used to achieve this. Both are used in symmetric cryptosystems that encrypt by blocks such as DES, Triple-DES, or AES, the proposal presented in this research work falls into that category, which is why they are also included.

In the proposed scheme, different permutations are performed in each execution. To build them requires a pseudo-random number obtained from Equation (13) in which it is proven that chaos has occurred. It is important to mention that for this task, only the digits to the right of the decimal point are taken into account. Then, a bijective function described in [22] is used; this algorithm adjusts to construct a permutation of $N$ elements depending on the dimensions $m \times n$ of the input image.

The same algorithm is used to build the S-Box; however, their design is a more complex task, since it must satisfy various criteria to be considered safe and resistant to DPA (differential power analysis) attacks that use statistical techniques to obtain information that helps to infer the encryption key of a cryptosystem and are usually more efficient than linear and differential attacks [26].

Perhaps the most important parameter to consider is non-linearity, which is the number of bits that must be modified in the truth table of the Boolean function to get closer to the closest affine function [27]. The non-linearity is represented in $GF(2^N)$, that is, $N = 2^{m-1} - 2^{\frac{m}{2}-1}$, for $m = 8$; thus, in theory, the upper bound of $N$ is 120 [28]. The S-Box proposed in this research work has a non-linearity of 100 equivalent to 78%, whose value is within the range of expected parameters [29], and it is measured using the Walsh function [22], which is presented in Equation (15).

$$LN = \min NL_i, \tag{15}$$

where

$$NL_i = 2^7 - \frac{1}{2} max_{a \in \mathbb{Z}_2^8} \left| W_{f_i}(a) \right|. \tag{16}$$

Table 2 shows a complete list of the cryptographic properties of the proposed S-Box, and they are compared with those of AES. Although our proposal only exceeds the second

S-Box in two parameters—Robustness to Differential Cryptanalysis and Transparency order—the others are found within the ranges of expected values for which it is considered adequate for the research being presented and robust enough against DPA attacks.

**Table 2.** Comparison of cryptographic properties between the proposed S-Box and AES S-Box.

| Cryptographic Properties | Proposed S-Box | AES S-Box | Expected Value (EV) [28] |
|---|---|---|---|
| Algebraic Degree | 7 | 7 | $EV \geq 10$ |
| Algebraic Immunity | 4 | 4 | $EV \leq 4$ |
| Balance | 0 | 0 | $EV = 0$ |
| Confusion Coefficient Variance | 0.142888 | 0.111 | $EV \leq 0$ |
| Correlation Immunity | 0 | 0 | $EV \leq 0$ |
| Nonlinearity | 100 | 112 | $100 \leq EV \leq 120$ |
| Fixed and Opposite Fixed Points | 0 | 0 | $EV \leq 4$ |
| Propagation Characteristic | 0 | 0 | $EV \leq 0$ |
| Robustness to Differential Cryptanalysis | 0.961 | 0.984 | $EV < 0.98$ |
| Signal to Noise Ratio (SNR) Diff. Power Analysis | 8.833 | 9.600 | $EV > 0.98$ |
| Transparency Order | 7.801 | 7.860 | $EV < 7.8$ |

The input parameters used to build the substitution box are listed below. In addition, the proposed S-Box and its inverse are shown in Figure 2.

(**a**)

(**b**)

**Figure 2.** (**a**) Proposed S-Box and (**b**) Inverse of proposed S-Box.

### 3.3. Complete Cryptosystem

This cryptosystem is symmetric; it is of the Substitution Permutation Network type, since the image encryption process is carried out through several encryption rounds, using XOR bitwise operations, a permutation, and the substitution box proposed in Section 3.2 [25]. Symmetric cryptosystems usually perform block encryption as with Triple-DES or AES [30,31], whose characteristic consists of dividing the information into blocks of fixed size (64 or 128). However, in this proposal, the encryption is carried out in one whose size is determined by the dimensions of the image, as already explained in Section 3.1. It was also explained that using a solution point of an Elliptic Curve, 15 encryption keys are generated, which will be applied one at a time for each of the 14 rounds of this proposal, except for the last one, which needs two. Starting from the fact that rounds 1 and 14 are the only different ones and rounds 2 to 13 are identical, we proceed to explain the complete operation of the algorithm. To describe the encryption and decryption process, the following nomenclature is used:

- $I$ for the input image.
- $IC$ for the encrypted image.
- $ID$ for the decrypted image.
- $R_{A,\ B}$, for each round, where $A$ is equal to the round number (1–14), and, $B$ for each intermediate operation having a maximum of 3.
- $O_A$, which denotes the output obtained after round $A$.
- $k_1$ to $k_{15}$ for every encryption key from one to 15.
- $P\ (R_{A,\ B})$ and $P^{-1}\ (R_{A,\ B})$ for the permutations that modify the previous output.
- $S_{BOX}\ (R_{A,\ B})$ and $S_{BOXINV}\ (R_{A,\ B})$ for the replacement box that modifies the previous output.

#### 3.3.1. Encryption Process

Step 1. Reading and Modifying the input image. An input image $I$ of dimensions $m\ \times\ n$ is read. If $m = n = 512$, the dimension $m$ is made to grow by one whereby $m = 513$ and $n = 512$. Then, the pixel values of the new row from position 0 to 255 will match with every bit of the integer $r$ used to generate the main key; that is, if $r = 1110\ldots0101$, pp (pixel in position) 0 = 1, pp 1 = 1, pp 2 = 1, pp 3 = 0, $\ldots$ , pp 252 = 0, pp 253 = 1, pp 254 = 0, pp 255 = 1. It is important to mention that these additional 256 pixels can serve as a fingerprint that verifies the origin of the received image in the decryption process. The pixels from position 256 to the last one can be filled with pseudo-random values.

Step 2. Round one of encryption. The operations carried out are summarized as follows:

- $R_{1,\ 1} = I \oplus k_1$
- $R_{1,\ 2} = S_{BOX}\ (R_{1,\ 1})$
- $O_1 = P\ (R_{1,\ 1})$.

Step 3. Rounds two through thirteen of encryption. The operations carried out are summarized as follows, and since all these rounds are the same, only two are described:

- $R_{2,\ 1} = O_1 \oplus k_2$
- $O_2 = S_{BOX}\ (R_{2,\ 1})$.

Step 4. Round fourteen of encryption. The operations carried out are summarized as follows:

- $R_{14,\ 1} = O_{13} \oplus k_{14}$
- $R_{14,\ 2} = S_{BOX}\ (R_{14,\ 1})$
- $R_{14,\ 3} = P^{-1}\ (R_{14,\ 2})$
- $IC = R_{14,\ 3} \oplus k_{15}$.

#### 3.3.2. Decryption Process

Step 1. Round one of decryption. IC is read, and the operations carried out are summarized as follows:

- $R_{1, 1} = IC \oplus k_{15}$
- $R_{1, 2} = P(R_{1, 1})$
- $R_{1, 3} = S_{BOXINV}(R_{1, 2})$
- $O_1 = R_{1, 3} \oplus k_{14}$.

Step 2. Rounds two through thirteen of decryption. The operations carried out are summarized as follows, and since all these rounds are the same, only two are described:

- $R_{2, 1} = S_{BOXINV}(O_1)$
- $O_2 = R_{2, 1} \oplus R_{2, 1}$.

Step 3. Round fourteen of decryption. The operations carried out are summarized as follows:

- $R_{14, 1} = P^{-1}(O_{14})$
- $R_{14, 2} = S_{BOXINV}(R_{14, 1})$
- $ID = R_{14, 2} \oplus k_{15}$.

Step 4. Reading and Modifying the output image. $ID$ is read; thus, the last row of pixels is deleted; in this case, the final dimensions of $ID$ will be $m = 512$ and $n = 512$. Finally, it is easy to conclude that $ID = I$.

## 4. Experiments and Security Analysis

In this section, we carry out all the required experiments to prove if this cryptosystem is resistant against the main types of cryptanalysis and modifications. Another piece of essential important information is provided, too. To facilitate understanding, graphs and tables are added.

### 4.1. Images for Experiments

The proposed cryptosystem has undergone several tests to demonstrate its robustness to the differential, lineal, and differential attacks. To carry out each experiment, JPEG images of different sizes have been chosen; these are:

- Lena.jpeg, Baboon.jpeg, and Boat.jpeg, with dimensions of $512 \times 512$ pixels.
- Barbara.jpeg, with dimensions of $720 \times 576$.
- City.jpeg, with dimensions of $1280 \times 720$.
- House.jpeg, with dimensions of $1920 \times 1080$.
- Security.jpeg, with dimensions of $4900 \times 3464$ pixels.

The first four images are commonly used in peer-reviewed papers, and the others are proposed in this research. The desktop application in which the proposed algorithm was implemented in Java programming language and the BufferedImage library was used to support certain tasks such as the reading of the pixels of each image [32]. All pictures are shown in Figure 3.

### 4.2. Elliptic Curve to Generate the Encryption Keys

To perform all the experiments, the elliptic curve has been chosen with the following data:

$a = $ 1bdb5c33c01c799169c62dae71c4176c5d1b0d
$b = $ 1b1dbcab63715d81db85c8db589685e3b5f1ec
$k = $ 59196553ed14991dbb65913217a9e4cffd676c103fa9df60df6b4b1681806d3571a8bfba36a
$p = $ 5e74aaa3d208d1c5322ef4c2327fa7e23143aa8eab32b38e0dd1705f83c7cd9909626a3f039
$q = $ 179d2aa8f48234714c8bbd308c9fe9f88c50eb8285ae4ae46740e766125f66f46313fd78995
$r\alpha = $ 8e28e3a5cef3c15189a8b1d9a8d87a2ffc9c79146fde29b7984fd997ea4b2fbc
$x^0 = $ 742fae6e43b6b33e4d4b49672c2097c87bbcbdba4f7b28b8969e844f19f3941acf646d88ae
$y^0 = $ 55546d0927fc1d205516fbbe4ceb24ed2495e01adc421a814a5b404167a3674e8f2c15d9bf

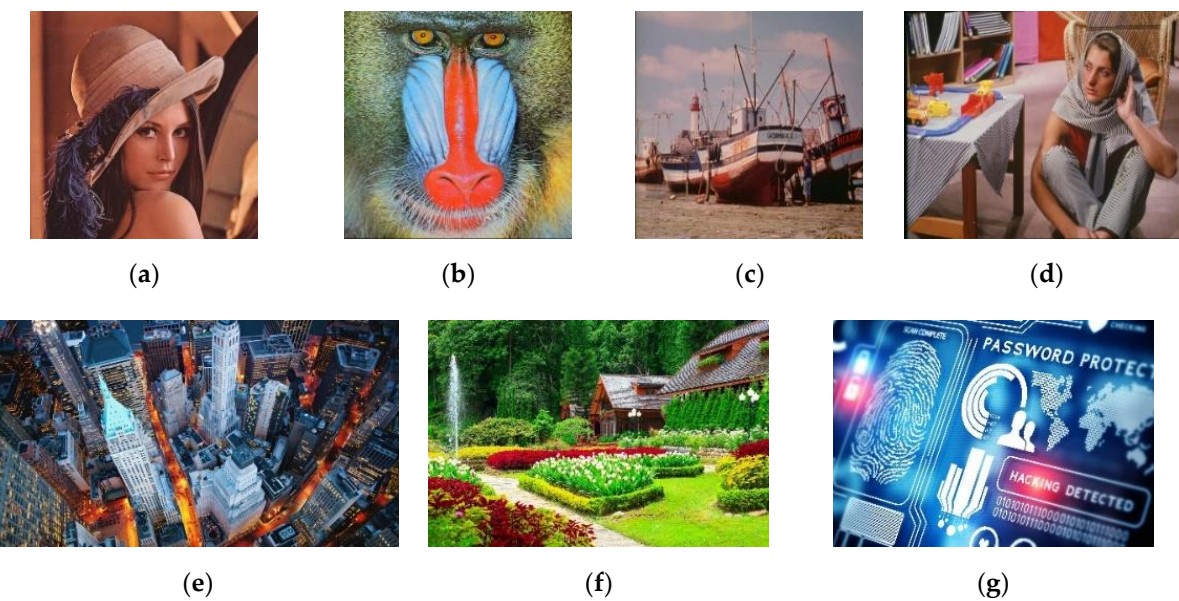

**Figure 3.** Images for tests: (**a**) Lena.jpeg, (**b**) Babbon.jpeg, (**c**) Boat.jpeg, (**d**) Barbara.jpeg, (**e**) City.jpeg, (**f**) House.jpeg, (**g**) Security.jpeg.

Next, fast analysis of the chosen elliptic curve is conducted:

- If the calculation $4(-k)^3 \bmod p$ is performed, the result obtained will be: 5a67ecc091bc 618e74d07f4905361dd6c7e67fc44d6d26c76fbfbd9625a8b43a4eddb4fee4a; therefore, it is shown that this is a Non-Singular Curve.
- Later, if the calculation $q \bmod p$ is performed, the result obtained will be: 179d2aa8f4823 4714c8bbd308c9fe9f88c50eb8285ae4ae46740e766125f66f46313fd78995; therefore, it is verified that this elliptic curve is not Supersingular.

On the other hand, it is easy to observe that $p$ and $q$ are different; in this way, it is stated that this elliptic curve is not of Trace One.

- Finally, this elliptic curve has a solution set $q$ of a size of $2^{256}$.

Thus, it is concluded that the chosen elliptic curve fulfills the four requirements described in Section 2.1 and is safe and suitable to generate the set of round keys.

### 4.3. Encrypted Images

Figure 4 shows the encrypted results of images shown in Figure 3. They prove that it is visibly impossible to find a pattern that allows inferring its origin. Nevertheless, it is important to understand that a visual inspection is not enough to demonstrate the cipher text is impossible to reverse; for that reason, in the next sections, several tests will be applied to ascertain it mathematically.

### 4.4. Statistical Cryptanalysis

An important aspect in any cryptosystem is to quantify the quality of the encryption, that is, the level of randomness, which will determine its resistance to statistical attacks that can determine the encryption key or the plaintext through existing biases or patterns in the ciphertext. Statistical tests such as those presented below are useful for this purpose.

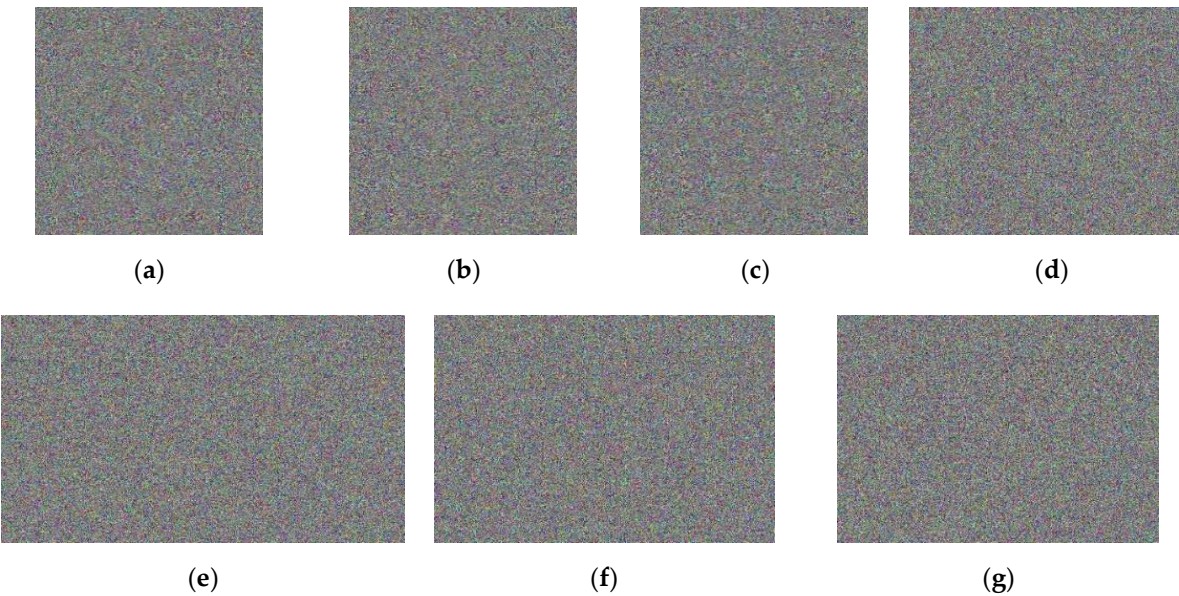

**Figure 4.** Images after encryption: (**a**) Lena_encrypted.jpeg, (**b**) Babbon_encrypted.jpeg, (**c**) Boat_ encrypted.jpeg, (**d**) Barbara_ encrypted.jpeg, (**e**) City_ encrypted.jpeg, (**f**) House_ encrypted.jpeg, (**g**) Security_ encrypted.jpeg.

### 4.4.1. Entropy

Entropy is one of the most important works of the French mathematician Claude E. Shannon [33]. In an image, it must be applied in each band of this depending on its type of color space. This test analyzes the histogram of the figure after it has been coded and determines if its frequency distribution is much more uniform than that of the original image. It is known that 255 is the maximum value of a pixel, and for its binary expression, it requires 8 bits; this implies that the perfect distribution of an image after being encrypted must be 8, which is unlikely to happen in practice. Therefore, any value greater than 7.9 indicates a high entropy level [15]. Equation (17) is used to calculate entropy.

$$H(x) = -\sum_{i=0}^{255} P(xi) log_2 P(xi) \qquad (17)$$

The results obtained in this test are shown in Table 3. The evaluation results show that the entropy provided by the proposed scheme is quite close to 8, which is the maximum theoretical value. Moreover, Figure 5 shows the histograms per every RGB channel of the Lena image used in the experiments to visualize its distribution before and after encryption. Here, it can be shown that after encryption, the estimated histogram is almost flat, independently of the histogram shape of the image before encryption. These types of graphs are widely used to carry out a visual inspection in this test.

**Table 3.** Entropy results.

| Image | Red Band | Green Band | Blue Band | Average Entropy |
|---|---|---|---|---|
| Lena | 7.9992825511 | 7.9993998215 | 7.9994452145 | 7.9993758623 |
| Baboon | 7.9994943826 | 7.9993538477 | 7.9993138633 | 7.9993873646 |
| Boat | 7.9992712402 | 7.9992818798 | 7.9992504561 | 7.9992678587 |
| Barbara | 7.9995083897 | 7.9995447744 | 7.9994407785 | 7.9994979809 |
| City | 7.9997759661 | 7.9997966970 | 7.9998150398 | 7.9997959010 |
| House | 7.9999254727 | 7.9999225925 | 7.9999318889 | 7.9999266513 |
| Security | 7.9999889277 | 7.9999888095 | 7.9999877460 | 7.9999884944 |

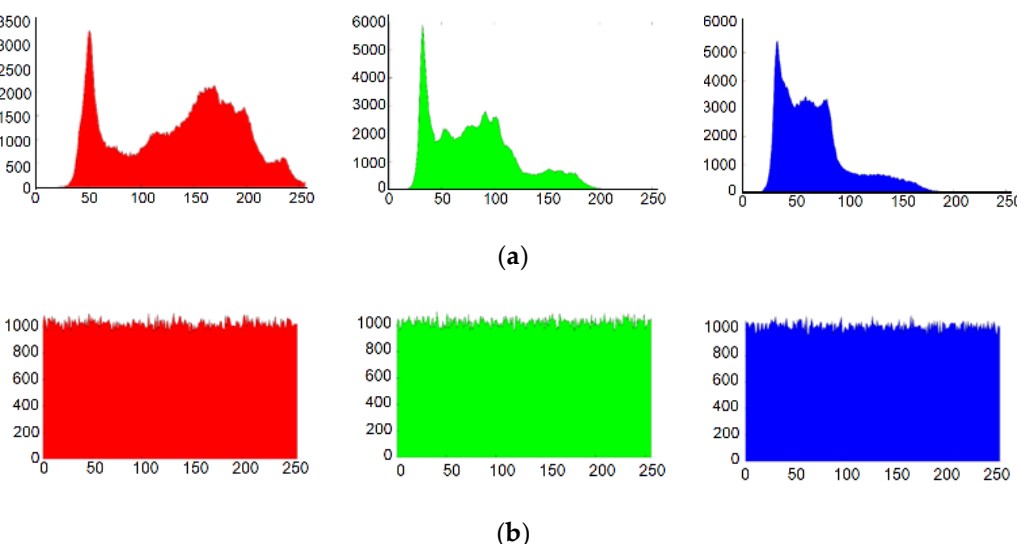

**Figure 5.** (**a**) Histograms of Lena.jpeg, (**b**) Histograms of Lena_encrypted.jpeg.

### 4.4.2. Correlation Coefficient

The correlation coefficient is based on the problem of analyzing the relationship between two variables $x$, $y$ [34]. In the specific case of image encryption, it analyzes whether the position of two contiguous pixels $x$, $y$ is determined by some given pattern or if there is a dependency between two pixels in each image. Otherwise, it is said that there is no correlation, and both are randomly positioned. This measure is obtained with Equation (18).

$$r = \frac{\sum[(x - \sum(x))(y - \sum(y)]}{\sqrt{\frac{i}{n}\sum_{i=1}^{n}(x_i - \sum(x))^2}\sqrt{\frac{i}{n}\sum_{i=1}^{n}(y_i - \sum(y))^2}} \tag{18}$$

If the measurement yields a 0, it is said that both images are completely different, and if a 1 or −1 is obtained, the conclusion would be that both images are equal. In practical terms, it is almost impossible to get zero, but anything value close to this indicates a high level of randomness between the pixels of an encrypted image. The most appropriate way to carry out this measurement is in three directions: Horizontal, that is, a pixel $x$ and its neighbour to the right; Vertical, that is, a pixel $x$ and its neighbor below; and Diagonal, that is, a pixel $x$ and its neighbor to below shifted by one space to the right.

Tables 4–6 show the correlation coefficients obtained; only absolute values are registered. The evaluation results show that the values of horizontal, vertical, and diagonal cross-correlation of the images under analysis closely approach zero, which means that it is not possible to infer one pixel of the image under analysis. Thus, if the correlation between the pixels of the encrypted image approach to zero, it is not possible to infer the original image using only information of the encrypted one. In addition to the experimental data given in Tables 2–6, Figure 6 shows the scatter plot per every RGB channel of one of the images used in the experiments to visualize the position of all the pixels before and after encryption; these types of graphs are widely used to carry out a visual inspection in this test. From this figure, it follows that in the encrypted image, it is not possible to estimate the value of pixel $(x, y)$ using the value of pixel on the other locations of the same image.

**Table 4.** Horizontal correlation coefficient results.

| Image | Red Band | Green Band | Blue Band | Average H.C.C. |
|---|---|---|---|---|
| Lena | 0.0027910911 | 0.0013075694 | 0.0012678227 | 0.0017888277 |
| Baboon | 0.0025146495 | 0.0014269271 | 0.0171631887 | 0.0070349218 |
| Boat | 0.0028113595 | 0.0108895696 | 0.0001350590 | 0.0046119960 |
| Barbara | 0.0011078457 | 0.0245272885 | 0.0010070337 | 0.0088807226 |
| City | 0.0014378943 | 0.0105296873 | 0.0059316278 | 0.0059664031 |
| House | 0.0013707650 | 0.0026946336 | 0.0082813835 | 0.0041155940 |
| Security | 0.0008586266 | 0.0050112873 | 0.0053137804 | 0.0037278981 |

**Table 5.** Vertical correlation coefficient results.

| Image | Red Band | Green Band | Blue Band | Average V.C.C. |
|---|---|---|---|---|
| Lena | 0.0043227617 | 0.0195862890 | 0.0014325272 | 0.0084471926 |
| Baboon | 0.0303389483 | 0.0029145088 | 0.0026712657 | 0.0119749076 |
| Boat | 0.0092702397 | 0.0061060780 | 0.0168940017 | 0.0107567731 |
| Barbara | 0.0027481028 | 0.0005638556 | 0.0046497850 | 0.0026539145 |
| City | 0.0212511418 | 0.0037570919 | 0.0013735665 | 0.0087939334 |
| House | 0.0064618621 | 0.0190488796 | 0.0005557270 | 0.0086888229 |
| Security | 0.0035772513 | 0.0052616476 | 0.0034203455 | 0.0040864148 |

**Table 6.** Diagonal correlation coefficient results.

| Image | Red Band | Green Band | Blue Band | Average D.C.C. |
|---|---|---|---|---|
| Lena | 0.0040161097 | 0.0081656937 | 0.0028642775 | 0.0050153603 |
| Baboon | 0.0061849334 | 0.0176082477 | 0.0021236495 | 0.0086389435 |
| Boat | 0.0076969924 | 0.0019718301 | 0.0062619588 | 0.0053102604 |
| Barbara | 0.0067001226 | 0.0017590721 | 0.0059405103 | 0.0047999017 |
| City | 0.0062177386 | 0.0049087084 | 0.0053074783 | 0.0054779751 |
| House | 0.0031439867 | 0.0043849522 | 0.0000507467 | 0.0025265619 |
| Security | 0.0098461466 | 0.0016037919 | 0.0054311831 | 0.0056270405 |

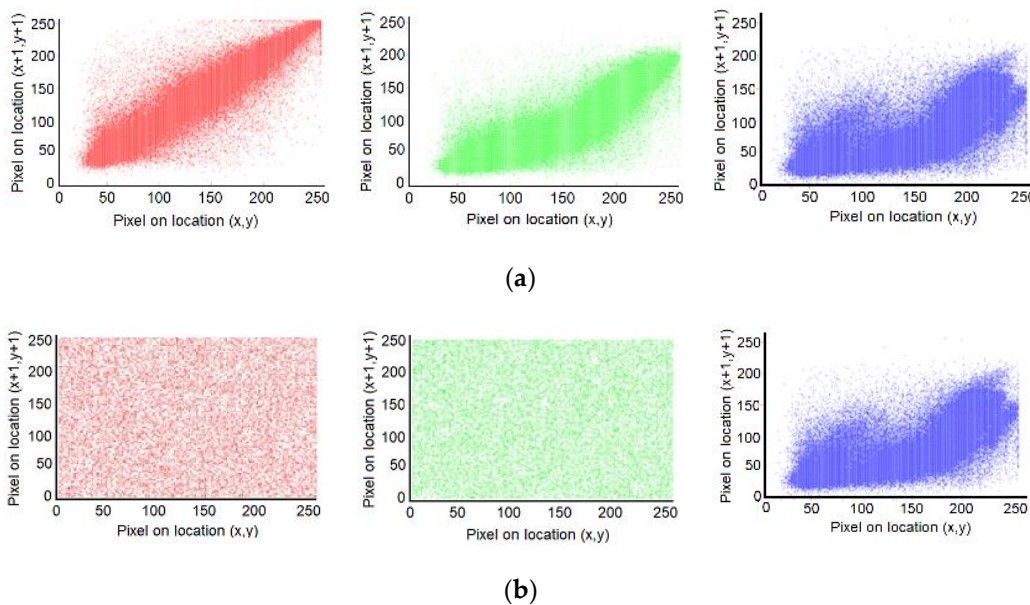

**Figure 6.** (**a**) Scatter plots of Lena.jpeg, (**b**) Scatter plots of Lena_encrypted.jpeg.

### 4.4.3. $\chi^2$ Test

This statistical test is based on the statement of two hypotheses: the first of them is known as null and is identified as $H_0$ [35]; in this case, its role is to affirm that the pixels of an image after being encrypted have a random distribution. The second one is known as the alternative and is called $H_1$; its function is to reject the assertion proposed by the null. Using Equation (19), $\chi^2$ is calculated, and it determines which of the two hypotheses is accepted and which is rejected. $f_o$ refers to the observed frequencies, that is, those of the encrypted image, and $f_e$ talks about the number of expected frequencies, which in this case is 256.

$$\chi^2 = \sum_{i=1}^{n} \frac{(fo_i - f_e)^2}{f_e}. \tag{19}$$

It is known that in tests based on hypotheses, there are two errors: a type I error that is the most important, that is, unequivocally rejecting, $H_0$; and, type II, that is, accepting $H_0$ wrongly. For this research work, the type I error is used, whose value is $\alpha = 0.01$. In practice, any threshold ($\chi^2$) less than 308 is enough for $H_0$ to be accepted [36]. Table 7 shows the results obtained from the test after the experiments.

**Table 7.** $\chi^2$ test results.

| Image | Red Band | Green Band | Blue Band | Average $\chi^2$ |
|---|---|---|---|---|
| Lena | 248.3114536 | 285.4836539 | 252.6187869 | 262.1379648 |
| Baboon | 184.2003975 | 234.2884898 | 249.1635853 | 222.5508242 |
| Boat | 264.0449749 | 261.5555760 | 272.5446485 | 266.0483998 |
| Barbara | 273.4035741 | 262.9516374 | 221.0672448 | 252.4741521 |
| City | 286.2245508 | 260.9730802 | 236.9778295 | 261.3918201 |
| House | 254.2933231 | 264.0568020 | 232.1029117 | 250.1510123 |
| Security | 260.6685332 | 263.5265517 | 188.4832084 | 237.5594311 |

### 4.5. Differential and Linear Cryptanalysis

Cryptanalysis is the antagonistic science of cryptography whose objective is to develop attacks capable of compromising or breaking encryption algorithms. Symmetric and block encryption cryptosystems must be robust to at least two types of cryptanalysis, differential and linear [37].

The differential attacks were first proposed by Eli Biham and Adi Shamir with the aim of breaking the DES cryptosystem [7]. Attacks of this type exploit the high probability of the existence of differences in the plain text $\Delta X$ and differences in the last round of encryption $\Delta Y$, which is known as the differential. This attack selects inputs and analyzes outputs to find the encryption key.

The linear attack was proposed for the first time by Mitsuru Matsui, who sought to exploit the DES cryptosystem through the known plaintext at entry [6]. This attack works considering the linear correlations between some of the plaintext bits (input block) and the output bits (cipher block) to infer the cipher key. There are various tests that a cryptosystem must undergo to verify that it is not vulnerable to the attacks described above; this process is carried out below.

### 4.5.1. NPCR and UACI

Both standards serve to test the resistance of any cryptosystem against differential cryptanalysis; they function as follows. Starting from the fact that there are two encrypted images, $C^1$ and $C^2$, which come from two images whose only difference is a pixel, and the encryption process has been carried out with the same keys. If the proposed cryptosystem is robust, images $C^1$ and $C^2$ must be practically different, which can be measured with the NPCR and UACI standards [38]. The first is defined by Equation (20) and the second is

defined by (21) where, $C^1$ and $C^2$ refer to the images, $T$ refers to the number of pixels in each image ($n \times m \times$ number of planes), and $D$ is defined in Equation (22).

$$NPCR : N\left(C^1, C^2\right) = \sum_{i,j} \frac{D(i,j)}{T} \times 100\%, \tag{20}$$

$$UACI : U\left(C^1, C^2\right) = \sum_{i,j} \frac{|C^1(i,j) - C^2(i,j)|}{255 * T} \times 100\%, \tag{21}$$

$$D(i,j) = \begin{cases} 0, \; if \; C^1(i,j) = C^2(i,j), \\ 1, \; if \; C^1(i,j) \neq C^2(i,j). \end{cases} \tag{22}$$

In the case of this experiment, both images are encrypted with the set of encryption keys, which are generated from the $K$ corresponding to the SHA-1 of both. In all the images used, pixel 3750 of the blue channel has been modified. In practical terms, in these tests, it is expected to obtain percentages between the range of 99.5% and 99.6% for the NPCR and between 33.4% and 33.5% for the UACI. The results obtained in this test are shown in Table 8.

**Table 8.** NPCR and UACI results.

| Image | NPCR (%) | UACI (%) |
| --- | --- | --- |
| Lena | 99.6034102908 | 33.4703504432 |
| Baboon | 99.6178778834 | 33.4530838915 |
| Boat | 99.6092480913 | 33.4670956081 |
| Barbara | 99.6141440401 | 33.4441091361 |
| City | 99.6236780513 | 33.4647289134 |
| House | 99.6113469295 | 33.4587882284 |
| Security | 99.6125747269 | 33.4640385760 |

### 4.5.2. Avalanche Effect

The avalanche effect, also known as avalanche attack, is another important manner to test the resistance of any cryptosystem against differential attacks [39]. This standard works with the same principle that was observed in NPCR and UACI tests, where a tiny change made in an image will produce a practically different ciphered image, although in this case, the change is made on a bit level. Having said that, let $I^1$ and $I^2$ be two plain images with only one different bit; then, they are ciphered using a group of round keys generated from a main key $K$ with just a distinct bit among each other. Thus, the resulting images $C^1$ and $C^2$ must be shown a bit rate of changing approached to 50% [40]. Using Equations (22) and (23), it is possible to obtain such a measure.

$$Avalanche = \sum_{i,j} \frac{D(i,j)}{Total\ bits} \times 100\% \tag{23}$$

The results obtained in the avalanche test are given in Table 9.

The evaluation results show that the avalanche effect resulting from the proposed scheme is quite close to the ideal value, which is equal to 50%.

**Table 9.** Avalanche effect result.

| Image | Avalanche Effect (%) |
| --- | --- |
| Lena | 49.9823570251 |
| Baboon | 50.0461260477 |
| Boat | 49.9830722808 |
| Barbara | 50.0082585841 |
| City | 49.9927435980 |
| House | 50.0056389702 |
| Security | 50.0402729690 |

### 4.5.3. Chosen/Known Plain-Text Attacks

These types of attacks fit into the category of linear cryptanalysis, and there four of them in total, which are the known-plaintext attack, the chosen-plaintext attack, the ciphertext-only attack, and the chosen-ciphertext attack; the first two are the most important, and it is stated that any cryptosystem capable of supporting them will do the same with the last two [39]. The procedure to test if a cryptosystem can resist them consists of encrypting two images: one white and one black, and then measuring their entropies and correlation coefficients and verifying if they fall within the parameters described in Sections 4.4.1 and 4.4.2. In this experiment, five images of each are used, corresponding to the dimensions of those chosen for the experimentation stage. Table 10 shows the average results obtained in this test.

**Table 10.** Average entropy and correlation coefficient results of a black and white image.

| Image | Dimensions | Average Entropy | Average H.C.C. | Average V.C.C. | Average D.C.C. |
|-------|-----------|----------------|----------------|----------------|----------------|
| White | 512 × 512 | 7.9993458278 | 0.0102043371 | 0.0217438421 | 0.0117693713 |
| Black | 512 × 512 | 7.9992852049 | 0.0177157935 | 0.0160536108 | 0.0088100118 |
| White | 720 × 576 | 7.9995995815 | 0.0078889236 | 0.0015165120 | 0.0051731527 |
| Black | 720 × 576 | 7.9995239552 | 0.0022354885 | 0.0079510811 | 0.0029753841 |
| White | 1280 × 720 | 7.9998072672 | 0.0021839664 | 0.0104461463 | 0.0246513849 |
| Black | 1280 × 720 | 7.9997950848 | 0.0013247086 | 0.0151392863 | 0.0018480579 |
| White | 1920 × 1080 | 7.9999026177 | 0.0129827377 | 0.0144222251 | 0.0066555182 |
| Black | 1920 × 1080 | 7.9999008370 | 0.0072195163 | 0.0187355224 | 0.0035342527 |
| White | 4900 × 3464 | 7.9999905211 | 0.0015279181 | 0.0048757459 | 0.0081609654 |
| Black | 4900 × 3464 | 7.9999889952 | 0.0028520019 | 0.0231656432 | 0.0013768905 |

### 4.6. Key Sensitivity Test

The objective of this test is to analyze the percentage of different pixels between two equal images, $C^1$ and $C^2$. Both will be encrypted with a set of encryption keys generated from a specific $K$, and in the case of the second image, the same $K$ will be used as for $C^1$ but with a small modification that could be the change of one bit. If we want to decrypt $C^1$ with the keys of $C^2$, it must be impossible, and vice versa. The percentage of change is calculated with Equation (24), and in practical terms, it is expected that minimum values of DiffImg equal to 99% will be obtained. Table 11 shows the results obtained from the test after the experiments.

$$DiffImg = \frac{DiffPixels}{TotalPixels} \times 100 \tag{24}$$

**Table 11.** Key sensitivity test results.

| Image | Different Pixels | Percentage Change |
|-------|-----------------|-------------------|
| Lena | 260,835 | 99.50065613% |
| Baboon | 260,903 | 99.52659607% |
| Boat | 260,517 | 99.37934875% |
| Barbara | 412,405 | 99.44179205% |
| City | 917,223 | 99.52506510% |
| House | 2,061,209 | 99.40244020% |
| Security | 16,892,441 | 99.52185158% |

### 4.7. Keyspace Analysis

The brute force attack on an encryption key is inescapable, where its objective is to try all possible combinations until finding the one that matches. In this case, the keys are generated from a solution point of an elliptic curve, which implies that there are $q$ different possibilities, because $q$ is a prime number. Furthermore, for a key to be secure, it must have a minimum size of $2^{100}$ bits. Since the keys depend on the integer $r$ which is 256 bits, the size of the keys used in this proposal is about $2^{256}$, which is far beyond the minimum required.

### 4.8. Occlusion and Noise Attacks

When an encrypted image is transmitted over an insecure medium, it takes the risk of being intentionally or incidentally modified or distorted by an attacker. Any clipping, obstruction, or change in pixels that can be interpreted as added noise implies information that has been lost and is impossible to recover. Nevertheless, if the image encryption quality of the image is high and all the pixels have been perfectly distributed in a way considered as random, when the image is decrypted, much of the original information will still be possible to display. Therefore, it is very important to measure the resistance of any cryptosystem, mainly to two attacks: that is the occlusion and the added noise.

Furthermore, in this test, an image that displays a written message is used, which is shown in Figure 7. Sometimes, these types of pictures are sent to share notes or news and, commonly, attackers can try to interfere or damage them to avoid that the communication can be completed.

Regarding the first attack, Lena.jpeg and Text.jpeg are encrypted, and then, various sections of an amplitude of 25%, 50%, and 75% of the total surface are intentionally cut out. Thus, when the image is deciphered, it is visually analyzed if the pixels remaining rearranged are enough to be able to infer which was the original image. The second attack consists of adding salt and pepper noise to the encrypted image, which in practice consists of scattering black or white pixels pseudo-randomly on the surface; for this case, the noise is added in densities of 25%, 50%, and 75%, and then, the same visual analysis is done as in the first one. The obtained results are shown in Figures 8–13.

"What We Know
Is A Drop,
What We Don't Know
Is An Ocean"

Issac Newton

**Figure 7.** Message.jpeg.

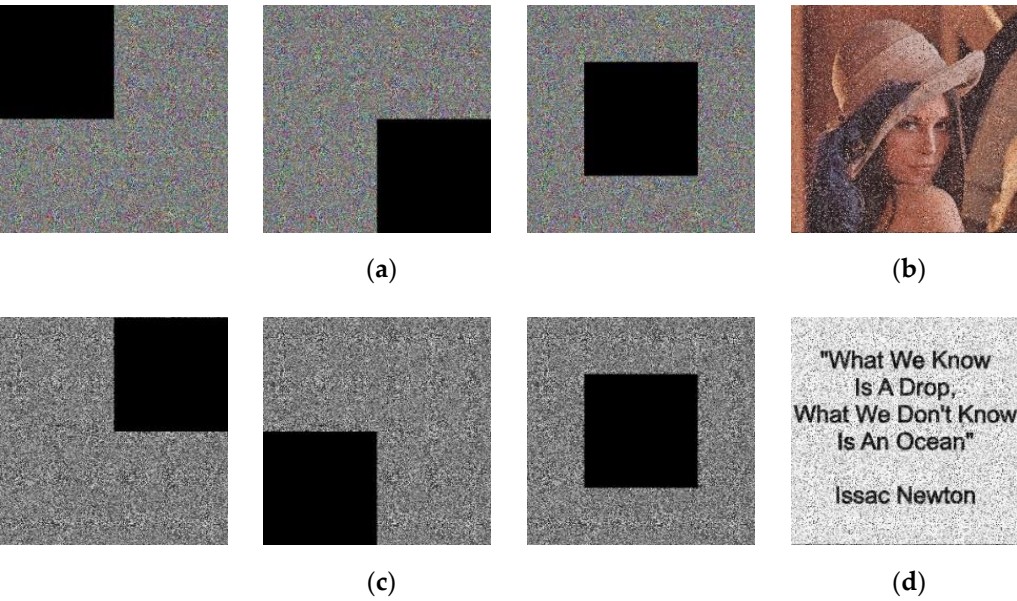

(a)     (b)

(c)     (d)

**Figure 8.** (**a**) Lena_encrypted.jpeg with 25% occlusion in different zones, (**b**) Decryption of Lena_encrypted.jpeg. after 25% occlusion attack, (**c**) Message_encrypted.jpeg with 25% occlusion in different zones, (**d**) Decryption of Message_encrypted.jpeg after 25% occlusion attack.

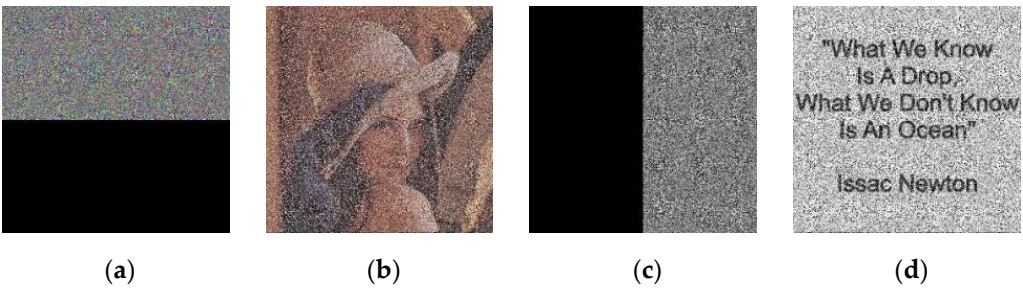

**Figure 9.** (**a**) Lena_encrypted.jpeg with 50% occlusion in different zones, (**b**) Decryption of Lena_encrypted.jpeg after 50% occlusion attack, (**c**) Message_encrypted.jpeg with 50% occlusion in different zones, (**d**) Decryption of Message_encrypted.jpeg after 50% occlusion attack.

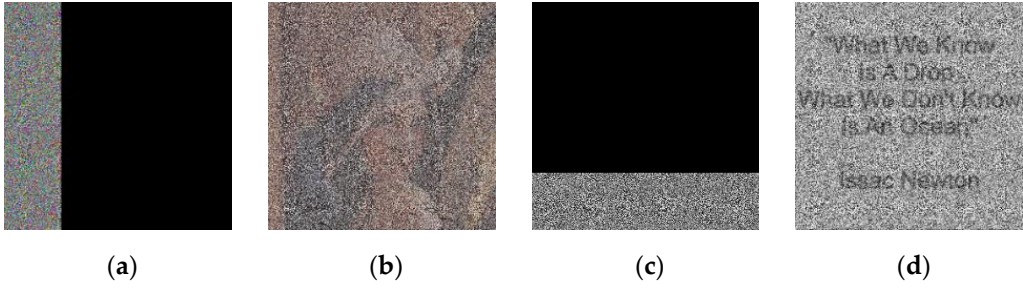

**Figure 10.** (**a**) Lena_encrypted.jpeg with 75% occlusion in different zones, (**b**) Decryption of Lena_encrypted.jpeg after 75% occlusion attack, (**c**) Message_encrypted.jpeg with 75% occlusion in different zones, (**d**) Decryption of Message_encrypted.jpeg after 75% occlusion attack.

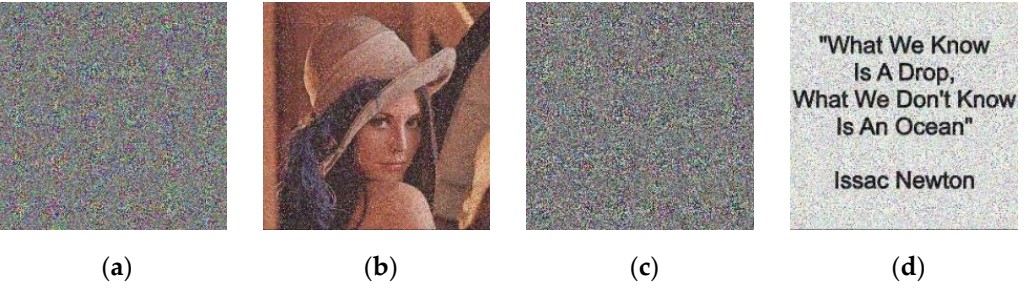

**Figure 11.** (**a**) Lena_encrypted.jpeg under a salt and pepper noise attack with a density of 25%, (**b**) Decryption of Lena_encrypted.jpeg after noise attack with a density of 25%, (**c**) Message_encrypted.jpeg under a salt and pepper noise attack with a density of 25%, (**d**) Decryption of Message_encrypted.jpeg after noise attack with a density of 25%.

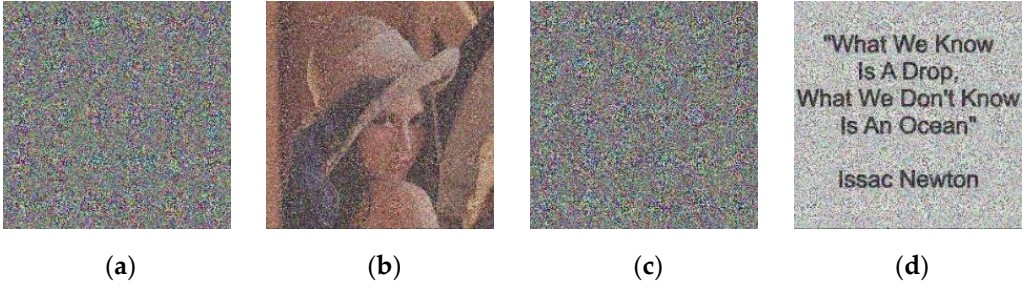

**Figure 12.** (**a**) Lena_encrypted.jpeg under a salt and pepper noise attack with a density of 50%, (**b**) Decryption of Lena_encrypted.jpeg after noise attack with a density of 50%, (**c**) Message_encrypted.jpeg under a salt and pepper noise attack with a density of 50%, (**d**) Decryption of Message_encrypted.jpeg after noise attack with a density of 50%.

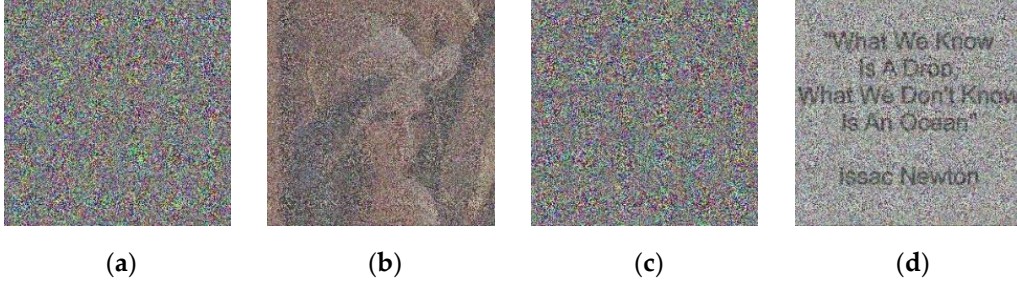

**Figure 13.** (**a**) Lena_encrypted.jpeg under a salt and pepper noise attack with a density of 75%, (**b**) Decryption of Lena_encrypted.jpeg after noise attack with a density of 75%, (**c**) Message_encrypted.jpeg under a salt and pepper noise attack with a density of 75%, (**d**) Decryption of Message_encrypted.jpeg after noise attack with a density of 75%.

*4.9. Time Encryption*

The computer used for all the tests had the next hardware resources:

- *Processor:* Intel Core I3 7350k, 4.00 GHz, dual-core.
- *RAM Memory:* 8 GB, 1600 Mhz, DDR3L.
- *Video Card:* Intel 630.
- *Hard disk:* SATA 7200 RPM.
- *Operative System*: Windows 10 Pro.

Having explained this, the speed encryption was measured five times for every image; thus, in Table 12 shows the average speed calculated.

**Table 12.** Average encryption time measured.

| Image | Dimensions | Encryption Time |
|-------|-----------|-----------------|
| Lena | 512 × 512 pixels | 0.123 s |
| Baboon | 512 × 512 pixels | 0.157 s |
| Boat | 512 × 512 pixels | 0.125 s |
| Barbara | 720 × 576 pixels | 0.189 s |
| City | 1280 × 720 pixels | 0.408 s |
| House | 1920 × 1080 pixels | 1.131 s |
| Security | 4900 × 3464 pixels | 7.946 s |

*4.10. Comparison with Other Articles of the State of the Art*

Table 13 shows the average results obtained in this research in four tests for entropy and correlation coefficient, while Table 14 shows a comparison of the NPCR, UACI, and the avalanche obtained by proposed scheme compared with those obtained using other similar papers recently published in the literature.

**Table 13.** Performance comparison between proposed scheme and others recently proposed.

| Peer-Reviewed Article | Entropy | H.C.C. | V.C.C. | D.C.C. |
|-----------------------|---------|--------|--------|--------|
| Ours-Lena | 7.999375 | 0.001788 | 0.008447 | 0.005015 |
| Ours-Baboon | 7.999387 | 0.007034 | 0.011974 | 0.008638 |
| Ref. [10]-Lena | 7.999300 | 0.001900 | 0.002400 | 0.001100 |
| Ref. [10]-Baboon | 7.999300 | 0.002400 | 0.001100 | 0.000800 |
| Ref. [11]-Lena | 7.999860 | 0.001500 | 0.000600 | 0.000900 |
| Ref. [11]-Baboon | 7.998840 | 0.001500 | 0.000640 | 0.000100 |
| Ref. [12]-Lena | 7.999200 | 0.003475 | 0.004013 | 0.007655 |
| Ref. [12]-Baboon | 7.999100 | 0.000380 | 0.005288 | 0.002308 |
| Ref. [13]-Lena | 7.991226 | 0.000100 | 0.008900 | 0.009100 |
| Ref. [14]-Lena | 7.996200 | 0.003110 | 0.054200 | 0.002310 |

**Table 14.** Performance comparison between proposed scheme and others recently proposed.

| Peer-Reviewed Article | NPCR | UACI | Avalanche |
|---|---|---|---|
| Ours-Lena | 99.6034 | 33.4703 | 49.9824 |
| Ours-Baboon | 99.6178 | 33.4530 | 50.0461 |
| Ref. [10]-Lena | 99.6113 | 33.4682 | 50.0334 |
| Ref. [10]-Baboon | 99.6112 | 33.4919 | 50.0411 |
| Ref. [11]-Lena | 99.6204 | 33.4898 | 50.0419 |
| Ref. [11]-Baboon | 99.6100 | 33.4600 | 50.0824 |
| Ref. [12]-Lena | —– | —– | 49.9322 |
| Ref. [12]-Baboon | —– | —– | 50.0324 |
| Ref. [13]-Lena | 100 | 33.4464 | —– |

Tables 13 and 14 shows that the evaluation results provided by the proposed are quite competitive with other previously proposed schemes.

## 5. Analysis and Results

This section provides an analysis of the evaluation results obtained by every test performed in a strictly sequential order as they were performed. The first evaluated parameter is the *Entropy*; according to the results of Section 4.4.1, this test seeks to find a value that means that the encrypted image has almost a uniform frequency distribution. If a value greater than 7.9 is obtained, the entropy is high and complies with the previous approach. It can be seen from Table 3 that the lowest calculated value is 7.9992, which is very close to the ideal value for which an image encoded with 8 bit/pixel is equal to 8. The next evaluated parameter is the Correlation coefficient. In Section 4.4.2, it was explained that this test seeks to measure the level of dependence between the contiguous pixels of an image encrypted in three directions: horizontal, vertical, and diagonal. There is a high non-linearity when the values approach zero, which happens in all cases. It should be remembered that the results of Tables 4–6 are expressed in absolute values. These tables show that the cross-correlation values of encrypted signals approach zero, which means that the knowledge of some pixel values does not allow the estimate the encrypted images. The $\chi^2$ *Test* proposed in Section 4.4.3 is based on the proposition of two hypotheses; the first indicates that the encryption carried out is random, and the second contradicts it; the acceptance of each one depends on a threshold $(\chi^2)$ that must be less than 308. It is observed that all the values recorded in Table 7 are less than 300 in all cases.

The resistance to differential and linear cryptoanalysis was also evaluated. To this end, the *NPCR and UACI* were estimated. As mentioned in Section 4.5.1, this test seeks to measure using both standards of the quantity and percentage of different pixels between two images whose only difference is one pixel. It was also explained that for the NPCR, minimum values of 99.5% are expected, and for the UACI, we expect values not less than 33.4%; as seen in Table 8, the lowest value of the first case is 99.60%, and for the second, it is 33.44%. Other test is the avalanche test, in which is evaluated that a tiny change made in an image will produce a practically different ciphered image. The evaluation results shown in Table 9 show that the obtained results are very close to 50%, which is the ideal value. The *Chosen/Known plain-text attacks* are the other recommended evaluation. In this test, black and white images of different dimensions are coded, and subsequently, their entropies and correlation coefficients are measured, which are recorded in Table 10. All measurements obtained are within the ranges specified in Sections 4.5.1 and 4.5.2.

The *Keyspace and key sensitivity* was also evaluated. In the case of the first, it was already explained in Section 4.7 that the minimum size for a key to be considered secure is $2^{100}$, which is far exceeded, since in our proposal, it is $2^{256}$. On the other hand, the second test measures the percentage of distinct pixels when an image is encrypted with two keys with minimal variation. Table 11 shows percentages greater than 99% that correspond to what is expected in practice. Another evaluated attack is the *Occlusion and noise attacks*. The objective of both is to determine with a visual inspection how much information is

recovered from an encrypted image that is covered or cropped to a certain extent as in the occlusion attack, or when it is randomly covered with white or black pixels after adding salt and pepper noise. For both cases, it is carried out in extensions or intensities of 25%, 50%, and 75%. Figures 8–13 show that although the level of visibility after decryption is lower with higher intensities, it is possible to perceive the message. Following the foregoing, it is determined that the results obtained are good and all the tests have been satisfactorily passed.

Finally, Tables 13 and 14 compare the results of the entropy, correlation coefficient, NPCI, UACI, and avalanche tests of the proposal presented with other articles of the state of the art. It is easy to see that the results obtained are close to those reported by other recently proposed papers. In some of the cases, they are equal or slightly lower, and in others, better measurements are obtained; therefore, it can be said that the system proposed in this research is competitive concerning the existing state of the art.

Finally, the future works that can be carried out for enhancing the performance of this research work are mentioned as follows. Researchers could develop substitution boxes with a higher level of non-linearity and better cryptography properties or use other logistic equations such as Lawrence's equation. The authors consider that it would also be of great value to verify the effectiveness of this proposal or others in the encryption of audio or video. Finally, the introduction of a steganography scheme could increase the security of the system and propose a scheme that includes a key distribution system.

## 6. Conclusions

In this article, an original symmetric cryptosystem has been presented whose purpose is the encryption of images in JPEG format, which stands out for its high quality, its lightness, and its ease of transmission, which makes it one of the most used at present. The contributions of this work that make it different from others that were mentioned in the state of the art are summarized as follows. Firstly, the use of elliptic curves with a constant $l$ equal to zero for the generation of the set of encryption keys. Next, an algorithm is proposed to generate the elliptic curves in addition to being mandatory that they comply with certain characteristics. The proposed scheme uses a chaotic logistic equation to generate permutations and a substitution box plus its inverse with a non-linearity level of 100. This algorithm also proposes the implementation of a fingerprint for the receiver to identify if the file received corresponds to the one sent by the issuer. The proposed algorithm carries out the encryption of the images in a single block, obtaining an adequate encryption speed. On the other hand, the cryptosystem has been subjected to several tests; after analyzing the results obtained, it is stated that this proposal is robust and capable of withstanding linear, differential, statistical, brute force, or modification attacks, as well as some of the better known as the discrete logarithm and the MOV. The proposed scheme provides quite competitive results compared with other previously proposed schemes.

**Author Contributions:** Conceptualization, V.S.-G., R.F.-C., H.P.-M.; methodology and software, E.H.-D.; formal analysis, E.H.-D., V.S.-G.; investigation, H.P.-M.; data analysis R.F.-C., V.S.-G.; writing and original draft preparation, E.H.-D., H.P.-M.; review, V.S.-G., R.F.-C. All authors have read and agreed to the published version of the manuscript.

**Funding:** This research received no external funding.

**Acknowledgments:** The authors would like to thank the Instituto Politécnico Nacional (National Polytechnic Institute), ESIME Culhuacan, CIDETEC, and CONACyT for their support to develop this researching work.

**Conflicts of Interest:** The authors declare no conflict of interest.

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
