# Peer review of "JPEG Images Encryption Scheme Using Elliptic Curves and A New S-Box Generated by Chaos"

_electronics, doi:10.3390/electronics10040413_

Round 1
Reviewer 1 Report
In general, it is a good research paper.
The context is well organized and well written.
The topic itself is interesting, but English must be slightly improved.
The algorithms are reasonable which are also proved by simulations.
A list of References is comprehensive. However, it is better that Comparisons are made with existing algorithms, e.g. with those deterministic or chaotic ones.
Author Response
Thank you for your comments. Attending to your comments we include a comparison with other existing algorithms which are shown in Table 13 and 14. The evaluation that we include in these Tables are: Entropy, Horizontal, vertical and diagonal correlation in Table 13. In Table 14 we include we include NPCR, UACI and Avalanche. We also include an extensive analysis of the evaluation carried out for evaluation of proposed scheme.
Reviewer 2 Report
Paper structure and organization needs improvement. Few comments are as follow:
- Replace the wordings “In this document” by “in this study or in this research”.
- Figure numbers are shown as “Error! Reference source not found” throughout the article. Update Latex properly.
- Both the Introduction and conclusion sections are poorly written. The contribution points given in the conclusions must be in the introduction part, and conclusions should have metadata and a summary of the findings.
- The points given for future work are also not presented well, and most are wrong. As the image encryption is shown in this research article, there is no need to do the PSNR calculation over such kind of blind encryption in the future.
- Avalanche attack is the mandatory attack to be executed for newly proposed chaos systems and must be performed to show the strength of the proposed chaos system. Do the avalanche attack experiments to prove the strength of the proposed chaos system.
Improve the structure and quality of the paper by incorporating the above suggestions before resubmission. English revisions are also required.
For improving the results section, perform the Avalanche attack(mandatory).
How your proposed scheme has significant improvements over the references [10, 11,12]? Do a clear discussion by considering other paraments too. Time complexity comparison is not sufficient.
Add Limitations of the Study section before conclusions and include future directions here with your work limitations.
Author Response
Thank you for your comments. Attending to your comments, we modify the paper as follows:
- The word document was replaced by paper.
- The paper was revised to avoid the legend (error Reference source not found or equation not found.
- Attending to your requirements, to introduction and conclusion were modified. In the introduction we include a description of the contribution of our paper.
- Attending to your requirements the future works were modified and the PSNR was eliminated.
- The avalanche evaluation was included in both Table 9 and Table 14.
- The grammatical errors were corrected. The English usage improved. We expect that in its actual form the paper be easier to understand.
- As mentioned above the avalanche test was performed and the obtained results compared with other previously proposed schemes.
- Attending to your comments we include a comparison with other existing algorithms which are shown in Table 13 and 14. The evaluation that we include in these Tables are: Entropy, Horizontal, vertical and diagonal correlation in Table 13. In Table 14 we include we include NPCR, UACI and Avalanche.
- Attending to your requirements, we also include an extensive analysis of the evaluation carried out for evaluation of proposed scheme, together with the future directions for improving the proposed scheme.
- The conclusions were also modified.
Round 2
Reviewer 2 Report
Paper organization and quality is much improved than the previous version.
But still, there are issues of English usage. For example, in the first line of Analysis and Results, the English correction is required:
In this section provides an analysis of the evaluation.....
Remove the word "In" from the start.
The Paper is accepted in its present form, but I'll suggest scanning the complete paper again for English corrections. It will further improve the readability of this paper.
The second concern is to make the bullet points in the contribution para, which is recently added in the introduction section. Research Contributions should be presented in the point form rather than in a para.